# Poor WASH (Water, Sanitation, and Hygiene) Conditions Are Associated with Leprosy in North Gondar, Ethiopia

**DOI:** 10.3390/ijerph17176061

**Published:** 2020-08-20

**Authors:** Lisa E. Emerson, Puneet Anantharam, Feleke M. Yehuala, Kassahun D. Bilcha, Annisa B. Tesfaye, Jessica K. Fairley

**Affiliations:** 1Rollins School of Public Health, Emory University, Atlanta, GA 30322, USA; lisa.emerson@ufl.edu (L.E.E.); puneetanantharam@gmail.com (P.A.); 2College of Medicine and Health Sciences, University of Gondar, P.O. Box 196 Gondar, Ethiopia; mogesfeleke@gmail.com (F.M.Y.); annisabefekadu@gmail.com (A.B.T.); 3School of Medicine, Emory University, Atlanta, GA 30322, USA; kbilcha@emory.edu

**Keywords:** leprosy, NTD, WASH, water, sanitation, hygiene, Ethiopia

## Abstract

Access to safe water, sanitation, and hygiene (WASH) is critical for preventing the spread of neglected tropical diseases (NTDs) including leprosy. WASH-related transmission factors remain largely unexplored in the leprosy transmission cycle. The aim of this project is to better understand WASH exposures among leprosy cases through a case-control study in North Gondar, Ethiopia. We hypothesized that leprosy cases were more likely to have inadequate WASH access and were more likely to have concurrent schistosomiasis, as schistosomiasis immune consequences may facilitate leprosy infection. Forty leprosy cases (forty-one controls) were enrolled, tested for *Schistosoma*
*mansoni*, administered a demographic and WASH survey, and assigned a WASH index score. WASH factors significantly associated with leprosy on adjusted analyses included open defecation (aOR = 19.9, 95% CI 2.2, 176.3) and lack of access to soap (aOR = 7.3, 95% CI 1.1, 49.9). *S. mansoni* was detected in 26% of participants and in stratified analysis those with leprosy had a 3.6 (95% CI (0.8, 15.9)) greater odds of schistosomiasis in districts bordering the lake, compared to 0.33 lower odds of schistosomiasis in districts not bordering the lake (95% CI (0.09, 1.2)). Overall, results suggest that leprosy transmission may be related to WASH adequacy and access as well as to schistosomiasis co-infection.

## 1. Introduction

Neglected tropical diseases (NTDs) infect over two billion of the world’s poorest people and disproportionately burden low- and middle-income countries (LMIC), which have a higher likelihood of poor water, sanitation, and hygiene (WASH). Treatment and control of NTDs is included in Goal 3 of the United Nation’s Sustainable Development Goals [1].

Leprosy, commonly known as Hansen’s disease, is a chronic infectious disease caused by *Mycobacterium leprae*, and is classified as an NTD. An active leprosy infection causes deforming skin lesions and permanent peripheral neuropathy and physical deformity if it is not treated early. Despite multidrug therapy (MDT) and recent public health interventions, close to 200,000 new leprosy cases are reported yearly with 14 LMICs reporting over 94% of all new disease [2]. While new leprosy cases in Ethiopia have declined each year, it remains a public health problem. In 2016, Ethiopia reported 3692 new leprosy cases and an incidence of 3.52 per 100,000 people [3].

*M. leprae* is associated with a complex immune response that differs based on type of leprosy: multibacillary (MB) and paucibacillary (PB). MB versus PB leprosy case types are classified by the number of lesions and results of skin smear tests [4]. MB leprosy is associated with a weakened Th1 immune response and an upregulation of Th2 mediated cytokines and inflammatory markers. PB leprosy is accompanied by a strong Th1 immune response. The majority of transmission is from MB leprosy [2].

The same causative agent, *M. leprae*, causes PB and MB leprosy types, but the type of leprosy a person develops appears to be dependent on the individual’s immune system’s response to the bacteria. Innate immune genetic variability in toll-like receptor polymorphisms is thought to be a factor related to the leprosy type a person exhibits. Indeed, recent studies estimate that close to 95% of the world’s population is not susceptible to leprosy [5]. Leprosy is thought to be transmitted primarily through nasal secretions or skin lesions of infected individuals; however, recent evidence suggests that zoonotic reservoirs, trauma-related skin-to-skin transmission, and environmental reservoirs may exist.

Schistosomiasis is another NTD caused primarily by three Schistosoma species, *Schistosoma mansoni*, *S. haematobium*, and *S. japonicum*. Schistosoma are transmitted through cercariae, which enter through the skin when a human comes into contact with water contaminated by human waste [6]. Recent studies have suggested that soil-transmitted helminth infection may facilitate or increase the risk of leprosy infection [7]. Helminth infections typically up-regulate the Th2 immune response and down-regulate the Th1 immune response, meaning the diminished Th1 response may lead to a lesser likelihood of controlling *M. leprae* infection, and therefore a higher likelihood of leprosy, in particular, MB leprosy [8]. A geographic association of overlapping schistosomiasis and leprosy was found in a co-endemic area of Brazil [7]. In Ethiopia, schistosomiasis infections affect close to five million persons and close to 4000 new leprosy cases are diagnosed each year [9]. Due to overlapping endemicity, this makes Ethiopia a good candidate to further study associations between the two infections.

Environmental factors and exposure through poor WASH conditions are associated with several NTDs including schistosomiasis, trachoma, and soil-transmitted helminths [10]. Despite several studies that have detected potentially viable *M. leprae* in water and soil samples and established survivability of *M. leprae* in soil, leprosy is largely ignored as a water or soil associated infection [11,12,13]. Even the recent WHO 2016-2020 Global Leprosy Strategy fails to mention water quality, quantity, and access as a tool for managing or preventing disease [14].

This study investigated the strength of association of WASH factors with leprosy infection through an unmatched case control study and further explored schistosomiasis-leprosy co-infections in the Amhara Region of Ethiopia. We hypothesized that poor WASH factors would be associated with a higher odds of leprosy infection and that there would be higher odds of having leprosy among subjects with schistosomiasis.

## 2. Materials and Methods

### 2.1. Study Population

A case-control study was conducted in May–October 2018 in the North Gondar, South Gondar, and Gondar Zuria Zones of the Amhara Region of Ethiopia, an area of 171,000 square kilometers, with a population of approximately 17 million people (50.2% male and 49.8% female) with endemic schistosomiasis and leprosy [15]. Study area is depicted in Figure 1. Leprosy cases were identified by a convenience sample via local leprosy registries and recruited at associated dermatology and family health clinics. Cases were defined as adults 18 and older with a clinical leprosy diagnosis presenting to Gondar area health district offices. Cases were limited to leprosy patients who were currently undergoing treatment or had been diagnosed within the last year. Pregnant women, unconfirmed cases, and those who had finished treatment were excluded as cases. Controls were selected from patients present at health offices without suspected leprosy (convenience sample), without previous leprosy infection, and who did not have close contact with a suspected or confirmed leprosy case. Exclusion criteria included children under the age of 18 and pregnant women. Disability grade, leprosy type, diagnosis date, and treatment medications were recorded for each case.

### 2.2. WASH Survey

A WASH survey with questions adapted from WHO/UNICEF Joint Monitoring Programme for Water Supply and Sanitation (JMP) core questions on water, sanitation, and hygiene for household surveys [17] (Appendix A) was administered to cases and controls to assess household WASH factors. These factors included water source (improved or unimproved), premises’ access to water, water treatment, time to fetch water, access to soap, handwashing practices, and sanitation facility use. The questionnaire also included relevant sociodemographic information such as age, gender, and education status.

### 2.3. Schistosomiasis Testing

Schistosomiasis testing was performed on all subjects using Schisto POC-CCA™ rapid test (Schisto POC-CCA cassette-based test; Rapid Medical Diagnostics, Pretoria, South Africa). These tests detect active *S. mansoni* infections in urine specimens. The sensitivity for the rapid test is 100% in intensities higher than 400 eggs per gram of feces and 70% in lower burden positive cases. The lowest detectable positive is with a worm burden of approximately 50 worms [18]. We performed the rapid test by collecting urine samples from participants and dropping 100 microliters of urine in the well on the cassette. Results were read after 20 min. Invalid tests were repeated.

### 2.4. WASH Index

A composite WASH index was created using the JMP service ladders. Scores were determined by classifying study participants’ level of service based on their answers to corresponding questions, as identified by the JMP core questions on water, sanitation, and hygiene for household surveys [17]. The index was scored on a scale of 0–5 for sanitation and drinking water and 0–3 for hygiene (Table 2) [19].

### 2.5. Statistical Analysis

Sample size was determined using the OpenEpi Sample Size for Unmatched Case Control calculator [20]. There are very few data to guide calculation of a sample size for the associations of WASH and leprosy. Since schistosomiasis is a WASH exposure and a variable in this study, we used published data on the associations of helminths and leprosy, which showed an odds ratio of four for the association between helminths and leprosy, to calculate sample size for this pilot study [7]. Based on an estimated schistosomiasis burden of 20–25% prevalence in North Gondar Zone, sample size was calculated using an alpha of 0.05 and power of 0.8, resulting in a goal of 40 cases and 40 controls. Data collected from the questionnaires was entered into SAS v. 9.4. Descriptive statistics were tabulated using frequencies, means, median, and standard deviations. Unadjusted odds ratio estimates were calculated first and then a logistic regression model was fit using WASH exposure factors including water source (improved or unimproved), premises’ access to water, water treatment, time to fetch water, access to soap, handwashing practices, sanitation facility use, and all potential confounders including sex, age, and education status. Sex and age have a known association with leprosy, typically with more leprosy found in men [5,21]. More leprosy is likely found in men due to social factors rather than underlying biological factors [22]. Education was included as a marker of socioeconomic status (SES) as some studies have shown an association with leprosy and poor SES [21,23]. Education was dichotomized as less than primary education and at least a primary education based on previous studies [21]. Stratified odds ratio estimates for leprosy-schistosomiasis co-infections were calculated after stratifying by proximity to Lake Tana. Woredas bordering Lake Tana were classified as close and woredas not bordering Lake Tana as far.

### 2.6. Human Subject and Ethical Considerations

Study participation was voluntary, without incentives or compensation for study participants. The study was approved by the Emory University (00103244) and University of Gondar (O/V/P/RCS/05/1467/2018) Institutional Review Boards. Informed verbal consent was given for all participants.

## 3. Results

Forty cases and 41 controls were recruited. Leprosy cases were predominantly male with majority MB disease (*n* = 83, %) and poor-moderate WASH access (Table 1). JMP service level scores for drinking water, sanitation, and hygiene are found in Table 2. Only one participant scored at the highest and safest tier of the sanitation ladder (tier 5). Thirty-three (41%) scored at the second lowest tier and 20 (25%) at the lowest tier. In other words, 53 (65%) participants used an unimproved sanitation source. Hygiene is a three-level ladder with 36 (44%) participants scoring at the top of the ladder with access to soap and water. This does not differentiate between types of handwashing facilities. Sixty-four (79%) participants had access to improved drinking water facilities, and 20 of these are premises’ access points.

In univariate analysis (Table 3), unimproved water source (OR = 4.22, 95% CI 1.07, 16.22), lack of premises’ water access (OR = 2.83, 95% CI 1.05, 7.65), lack of soap (OR = 2.61, 95% CI 1.06, 6.42), lack of handwashing (OR = 4.56, 95% CI 1.69, 12.28), and open defecation (OR = 4.32, 95% CI 1.67, 11.18) were associated with leprosy. Lack of water treatment, time to fetch water, schistosomiasis, and distance to the lake were not conclusively associated with leprosy infection. In adjusted analysis, open defecation (OR = 19.86, 95% CI 2.24, 176.25) and lack of access to soap (OR = 7.31, 95% CI 1.07, 49.94) were associated with leprosy. Water source and lack of handwashing had point estimates suggestive of an association with leprosy infection. Lack of water treatment and time to fetch water were not associated with leprosy infection.

Odds ratio estimates based on relation to Lake Tana are included in Table 4. Those with leprosy had greater odds of schistosomiasis in districts bordering Lake Tana (OR = 3.56, 95% CI 0.80, 15.85), while those with leprosy had lower odds (OR = 0.33, 95% CI 0.09, 1.19) of schistosomiasis in districts not bordering the lake; however, these were not statistically significant.

## 4. Discussion

Overall WASH index scores were low for participants, particularly on the sanitation and hygiene ladders and, as predicted, our study found an association between several WASH factors, including soap access and open defecation, with leprosy infection. We also hypothesized that there would be an association between schistosomiasis and leprosy based upon previous studies linking helminth infection to leprosy infections and the immune response to helminth infections [7,8]. Helminth infections up-regulate the Th2 immune response and down-regulate the Th1 response, which reduces the immune system’s ability to control *M. leprae* infection. However, overall, schistosomiasis infection was not found to be significantly related to leprosy infection, although a larger study may have detected a difference given the results of the stratified analysis which suggested that schistosomiasis may have been associated with leprosy in regions nearer to the lake, where schistosomiasis is more highly endemic.

WASH index scores in general were low for the entire study population but were especially low for subjects with leprosy. While most individuals with leprosy had access to an improved water source, only 20% had water access in their home or yard. In sanitation, more than one-third of cases did not have access to any facility and none had access to an improved facility not shared with other households. Finally, half of leprosy cases did not have access to soap and water for handwashing. Water source, lack of premises’ access to water, lack of soap, lack of handwashing, and open defecation were statistically correlated with leprosy cases upon unadjusted analyses. Adjusted analyses showed open defecation and lack of soap were correlated with leprosy cases. Overall, these results support a relationship between WASH factors and leprosy cases.

These results are thus important due to the burden of both poor WASH and leprosy in LMICs. In fact, fourteen LMICs report over 94% of the nearly 200,000 new leprosy cases every year [2]. While the transmission of new cases is thought to occur primarily through nasal secretions of infected individuals, recent evidence suggests that other transmission routes and factors, such as environmental reservoirs, should be considered [23]. This project thus explored the previously understudied associations of WASH factors and leprosy infection and identifies the need for further study, including the relationship between schistosomiasis and leprosy.

Recent studies have suggested environmental reservoirs of leprosy may exist [24,25]. Water that is shared or reused from a source patient may become environmental reservoirs for infection, possibly by aerosolization of *M. leprae*. Contaminated water may be capable of transmitting viable *M. leprae*, as recent studies have found potentially viable bacteria in water and in amoebas commonly found in untreated water [24,25]. The transmissibility of leprosy bacteria found in the environment is difficult to prove, however, as *M. leprae* does not grow in laboratory conditions [26]. However, experimentally based studies with mouse foot pad cultures do prove the survivability of *M. leprae* in soil and free-living amoeba [24]. Studies on environmental reservoirs of leprosy have relied primarily on reverse-transcriptase polymerase chain reaction (rt-PCR) of leprosy to assess reliability, and one study quantifying mRNA from *M. leprae* lends stronger support to the ability of these studies to assess viable bacteria in environmental samples [12,13,21,23,27].

While studies have linked water to possible routes of infection, sanitation has not been implicated as a potential transmission route of leprosy [23]. Potentially viable *M. leprae* has been found in soil samples, but the route of transfer from person to environment is not understood [12]. This is likely to be from shedding from the skin rather than stool, as viable *M. leprae* is not known to be excreted in stool [12]. Open defecation was associated with leprosy in this study, and we believe that open defecation is significant because it accounts for other factors including socio-economic status and potential exposure to soil-transmitted helminth infections which may then increase the risk of active leprosy [28].

Our analysis of region-specific schistosomiasis infection points towards an association between schistosomiasis and leprosy within the Lake Tana area. This association may be due to increased exposure to water with schistosomiasis present. While proximity to a body of water should be considered as a WASH factor and as an important contributor to schistosomiasis infection, we do not know if there is a true difference in schistosomiasis-leprosy co-infections between communities near a body of water and far from a body of water since our study was not powered to detect a statistically significant difference in schistosomiasis status by proximity to the lake.

Due to the similarity in WASH access to other populations in developing countries, the associations found in this project are likely to be generalizable to populations with a large number of residents living in poverty and communities with endemic leprosy and schistosomiasis. The findings point towards a general association between WASH and leprosy and, while not necessarily significant in all findings, express the need for further research on water as a transmission route and associations with WASH factors and leprosy.

Our study faced several key limitations including a small sample size, which limited the model reliability and the overall statistical analysis of the project. MB leprosy cases were more likely to be detected in our study, potentially due to increased visibility of MB leprosy compared to PB. This could also have been observed due to an increase in Th2 responses, which can occur as a result of co-infection, poor nutrition, or other immune perturbations. Our results showed wide confidence intervals for several exposure variables due to the small sample size and similarity in survey responses among large proportions of study population. A further weakness of our study was the presence of unquantified confounders such as economic status of participants. Education was used to account for this factor in our study, but a more direct measure of SES could lead to better estimates. Other missing key confounders include potential presence of soil-transmitted helminth infections in addition to schistosomiasis. Our study is potentially limited by selection bias, which is common in case-control study designs due to sampling. Cases were selected from leprosy registries, and subjects on these registries may be more able to seek medical care, have adequate transportation, and more health education. Controls were selected from health offices and may be more likely than the target population to have underlying health conditions.

While this study does not prove a causal relationship between WASH factors and leprosy, it does elucidate the need for further study and analysis of these relationships between WASH and leprosy. Future studies should further consider WASH factors presented in this study and soil-transmitted helminth infections, water quantity, water quality, bathing frequency, water-reuse behaviors, and poverty-related factors as risks for leprosy. Future studies should also consider a larger sample size to study WASH factors and schistosomiasis infection as exposures for leprosy infection. Cohort studies may be considered to study schistosomiasis and WASH factors as exposure variables over time. Future studies can also be improved by active case-finding and visiting subjects at their residence rather than subjects’ health care centers.

## 5. Conclusions

In conclusion, our study found that WASH factors including water source, premises’ access to water, access to soap, handwashing practices, and open defection are related to leprosy infection. Schistosomiasis infection, however, was not found to be significantly associated with leprosy infection. Overall, this project supports the hypothesis that WASH conditions are associated with leprosy transmission and warrant further investigation. Further investigations with larger sample sizes will determine the extent of the association between leprosy and WASH factors.

## Figures and Tables

**Figure 1 ijerph-17-06061-f001:**
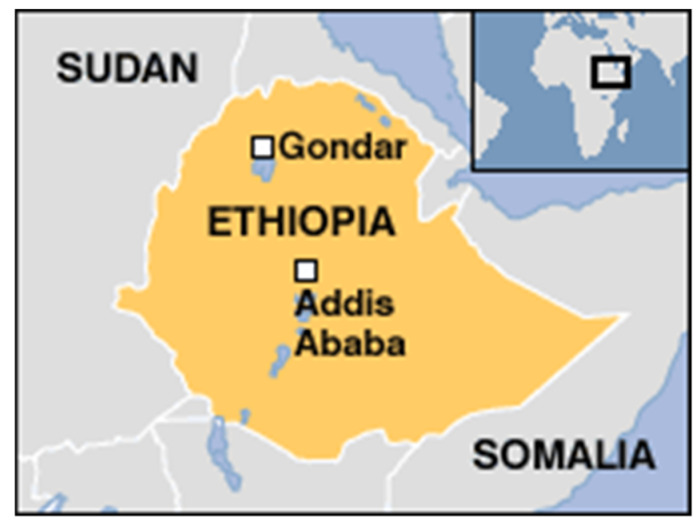
Map of Ethiopia with study area. Gondar is depicted on this map and location of Lake Tana can be seen as the blue lake just South of Gondar [16].

**Table 1 ijerph-17-06061-t001:** Demographic and Clinical Variables.

Variable	Total (*n* = 81)
Age, years (median, SD)	33 (18.4)
Sex, *n* (%)	
Male	61 (75.3)
Leprosy Type, *n* (%)	
Total Leprosy Cases	40
MB	33 (82.5)
PB	4 (10)
Unknown	3 (7.5)
Disability Grade at Diagnosis	
Total Leprosy Cases	40
Grade0	20 (50)
Grade1	12 (30)
Grade2	8 (5)
Positive Schistosomiasis, *n* (%)	21 (25.9)
Drinking Water Source, *n* (%)	
Piped Water to House/Yard	25 (30.9)
Public Tap/Standpipe	29 (35.8)
Other Protected Source	14 (17.3)
Unprotected Spring	10 (12.4)
Surface Water	3 (3.7)
Water fetching time in minutes (median, SD)	10 (15.4)
Toilet Facility, *n* (%)	
Flush Toilet	5 (6.2)
Improved Pit Latrine	23 (28.8)
Pit Latrine	32 (39.5)
No Toilet Facility	20 (24.7)
Household Water Treatment, *n* (%)	
Yes	21 (25.9)
No	60 (74)
Soap Available for Handwashing, *n* (%)	
Yes	44 (54.3)
No	36 (44.4)
Handwashing at home, *n* (%)	
No handwashing	29 (35.8)
In kitchen	14 (17.3)
By latrine/toilet	6 (7.4)

**Table 2 ijerph-17-06061-t002:** WASH Index Scores ^a^.

Drinking Water	Total *n* (%)	Leprosy Cases, *n* (%)	Sanitation	Total *n*, (%)	Leprosy Cases, *n* (%)	Hygiene ^b^	Total *n*, (%)	Leprosy Cases, *n* (%)
Tier 5. Drinking water from an improved water source, which is located on premises, available when needed and free from fecal and priority chemical contamination.	24 (30)	8 (20)	Tier 5. Use of improved facilities which are not shared with other households and where excreta are safely disposed in situ or transported and treated off-site.	1 (1)	0 (0)			
Tier 4. Drinking water from an improved source, provided collection time is not more than 30 min for a roundtrip including queuing.	40 (49)	22 (55)	Tier 4. Use of improved facilities, which are not shared with other households.	14 (17)	11 (28)	Tier 3. Availability of a handwashing facility on premises with soap and water.	36 (44)	11 (28)
Tier 3. Drinking water from an improved source for which collection time exceeds 30 min for a roundtrip including queuing.	3 (4)	0 (0)	Tier 3. Use of improved facilities shared between two or more households.	13 (16)	3 (8)	Tier 2. Availability of a handwashing facility on premises without soap and water.	16 (20)	8 (20)
Tier 2. Drinking water from an unprotected dug well or unprotected spring.	10 (12)	9 (23)	Tier 2. Use of pit latrines without a slab or platform, hanging latrines or bucket latrines.	33 (41)	11 (28)	Tier 1. No handwashing facility on premises.	29 (36)	20 (50)
Tier 1. Drinking water directly from a river, dam, lake pond, stream, canal or irrigation canal.	4 (5)	1 (3)	Tier 1. Disposal of human feces in fields, forests, bushes, open bodies of water, beaches, and other open spaces or with solid waste.	20 (25)	15 (38)			

^a^ Score ladder taken from Joint Monitoring Programme for Water Supply and Sanitation (JMP) service ladders. ^b^ Hygiene is a three level ladder while drinking water and sanitation are five level.

**Table 3 ijerph-17-06061-t003:** Unadjusted and Adjusted Water, Sanitation and Hygiene (WASH) Factor Associations.

Covariate	OR	Lower Limit ^a^	Upper Limit	aOR ^f^	Lower Limit ^a^	Upper Limit
Water Source ^b^	4.22	1.07	16.72	3.47	0.31	38.76
Lack of Premise Water Access ^c,d^	2.83	1.05	7.65	-	-	-
Lack of Water Treatment	1.62	0.59	4.40	0.28	0.04	1.79
Time to get water	1.01	0.98	1.04	0.99	0.93	1.05
Lack of Soap	2.61	1.06	6.42	7.31	1.07	49.94
Lack of Handwashing	4.56	1.69	12.28	2.46	0.47	12.82
Open Defecation	4.32	1.67	11.18	19.86	2.24	176.25
Schistosomiasis ^d^	0.54	0.20	1.49	-	-	-
Lake Region ^d,e^	0.60	0.22	1.69	-	-	-

^a^ 95% confidence limits; ^b^ Unimproved vs. Improved; ^c^ In-home or in-yard water access; ^d^ Not included in adjusted model; ^e^ Region bordering Lake Tana. Not included in adjusted model; ^f^ Model adjusted for age, sex, and education.

**Table 4 ijerph-17-06061-t004:** Leprosy-Schistosomiasis Co-infection Stratified by Proximity to Lake Tana.

Strata	*n* (%)	OR	Lower Limit ^a^	Upper Limit ^a^
Near Lake Tana	20	3.56	0.80	15.85
Distant from Lake Tana	61	0.33	0.09	1.19

^a^ 95% confidence limits.

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
