# Peer review of "Poor WASH (Water, Sanitation, and Hygiene) Conditions Are Associated with Leprosy in North Gondar, Ethiopia"

_ijerph, 2020, doi:10.3390/ijerph17176061_

Round 1
Reviewer 1 Report
General comment: The study is an attempt to connect WASH and leprosy which is relevant to the ongoing discussion to work jointly on NTDs, WASH and nutrition for better results and sustainability of the disease specific programmes. The attempt to identify the co-infection of leprosy and Schistosomiasis. However, there is a decent scope to improve the current study information.
Specific comment:
- Line 39 says person-years, whereas to my knowledge, [3] says incidence of new cases per 100,000 persons.
- Line 46 is also not correct, and PB may transmit disease.
- Line 73, i don't think that you used case-control and logistic regression just to prove an association. case-control is to track back the exposure from the outcome and logistic R tells the strength of association and can predict the future. For association, only hypothesis testing and chi-square would be sufficient. So kindly match your objectives with the methodology.
- In the method section please describe the population baseline for the parameters you collected. You only mentioned the population of that area inline 83 which is not sufficient to understand that the sample you draw was representative or how much different from the general population of that area. I suggest adding a column in table 1 to inform about the baseline as much as you can.
- line 130, leprosy is gender neutral, but more men are reported in some countries due to gender equality and programme selectivity. If you see the WHO leprosy weekly epid. report then there are some countries which also reports women comparably to men. Please correct.
- please provide a map of this area/catchment for better understanding.
- line 146 is poorly written and confusing. please correct the sentence structuring.
- There is no info on how you calculated the sample size or bases.
- Table 1. grade 0 and 1 data is not credible that's why WHO also does not report it.
- what is the unit of water fetching time in table 1?
- I would recommend to put current table 2 in the appendix and only presenting the tier/category wise score which you applied in the analysis (optional).
- Please improve table 2 presentation by possibly adding a line between DW, sanitation and Hygiene OR add a column to show clubbed ladders categories for analysis. In table 2 please add a column to present how you clubbed ladders for analysis. The write-up alone is not sufficient.
- Presentation of table 2 bottom can be improved by adding space between footnote and write-up
- The second footnote inline 155 is part of table 3, not 2
- remove schistosomiasis form table 3 analysis, because you want to see it as co-infection, not as a risk-factor or exposure. keep your co-infection analysis separate then exposure.
- Table 4 analysis should be stratified between cases and controls only in Near lake tana if you have some sample. It will be low of course, but you can mentioning posthoc analysis with a careful discussion that the sample size is not sufficient to claim result sound, but gives a certain idea for future studies to consider this aspect while calculating sample size considering co-infection endemicity.
- discuss that why you have such a broad CI for handwashing and open defecation. You mentioned low sample size, but connect the dots with high CI.
- Also, discuss the sample selection bias.
- line 148 "sanitation" should be "hygiene".
- Discuss why you found majority MB cases. Maybe because weak active case finding in the area which is leading to more severely infected cases/MB.
- Discuss the effect of sampling form local registry because they are programme generated and biased by the local socio-economics and programme operations.
- inline 202-213, you are not discussing your results. It sounds like background/intro.
- inline 214-221 your discussion is good.
- line 226-227 says your study was not designed for co-infection btwn water source, whereas it was your second objective. contradictory statement. you already categorised lep with distance so this is not valid.
- line 246-249 is a repetition. avoid and use space to conclude effectively. you have some good results to conclude, for example, results indicate that poor wash condition is a risk factor for leprosy though association should be further validated with a large sample size study. The Lep and schis co-infection still cannot be ruled out and need stratified analysis with sufficient sample size.
- Based on your results also write a recommendation for programmes in the discussion section.
Author Response
Dear IJERPH Editorial Office:
Thank you for giving us the opportunity to revise our draft of the manuscript titled “Poor WASH (Water, Sanitation, and Hygiene) Conditions are Associated with Leprosy in North Gondar, Ethiopia” for publication in the International Journal of Environmental Research and Public Health. We appreciate the reviewers’ feedback and their thoughtful input. We have incorporated most of the reviewers’ suggestions, which are highlighted in the manuscript. Please see our point-by-point response to the reviewers’ comments.
- Line 39 says person-years, whereas to my knowledge, [3] says incidence of new cases per 100,000 persons.
We have edited the paper accordingly (line 80).
- Line 46 is also not correct, and PB may transmit disease.
We have edited the paper accordingly (line 87)
- Line 73, i don't think that you used case-control and logistic regression just to prove an association. case-control is to track back the exposure from the outcome and logistic R tells the strength of association and can predict the future. For association, only hypothesis testing and chi-square would be sufficient. So kindly match your objectives with the methodology.
Logistic regression also measures association by accounting for confounders, as opposed to chi-sq which is a bivariate analyses. We have edited the paper to clarify that we are measuring the strength of the association (line 114) .
- In the method section please describe the population baseline for the parameters you collected. You only mentioned the population of that area inline 83 which is not sufficient to understand that the sample you draw was representative or how much different from the general population of that area. I suggest adding a column in table 1 to inform about the baseline as much as you can.
We have added more information regarding our study population and sampling strategy.
- line 130, leprosy is gender neutral, but more men are reported in some countries due to gender equality and programme selectivity. If you see the WHO leprosy weekly epid. report then there are some countries which also reports women comparably to men. Please correct.
We have edited accordingly.
- please provide a map of this area/catchment for better understanding.
We have added a map of this area.
- line 146 is poorly written and confusing. please correct the sentence structuring.
We have edited for clarity.
- There is no info on how you calculated the sample size or bases.
Sample size information has been added to methods section (see statistical analysis section in methods).
- Table 1. grade 0 and 1 data is not credible that's why WHO also does not report it.
We are unsure of what the reviewers are referring to and are reporting grades of disability that are defined by WHO.
- what is the unit of water fetching time in table 1?
Units have been added to table 1.
- I would recommend to put current table 2 in the appendix and only presenting the tier/category wise score which you applied in the analysis (optional).
Thank you for this insightful suggestion. We have decided to leave this table in the manuscript as it is vital information for understanding WASH adequacy and access of subjects.
- Please improve table 2 presentation by possibly adding a line between DW, sanitation and Hygiene OR add a column to show clubbed ladders categories for analysis. In table 2 please add a column to present how you clubbed ladders for analysis. The write-up alone is not sufficient.
We have edited this table accordingly.
- Presentation of table 2 bottom can be improved by adding space between footnote and write-up
We have edited this table accordingly.
- The second footnote inline 155 is part of table 3, not 2
We have ensured footnotes are referring to correct tables.
- remove schistosomiasis form table 3 analysis, because you want to see it as co-infection, not as a risk-factor or exposure. keep your co-infection analysis separate then exposure.
Schistosomiasis is an exposure for leprosy infection in our analysis, so it is treated like other exposure variables.
- Table 4 analysis should be stratified between cases and controls only in Near lake tana if you have some sample. It will be low of course, but you can mentioning posthoc analysis with a careful discussion that the sample size is not sufficient to claim result sound, but gives a certain idea for future studies to consider this aspect while calculating sample size considering co-infection endemicity.
We have added an n column for Table 4 to aid in understanding.
- discuss that why you have such a broad CI for handwashing and open defecation. You mentioned low sample size, but connect the dots with high CI.
We have edited accordingly in the discussion section.
- Also, discuss the sample selection bias.
We have highlighted this in the limitations section.
- line 148 "sanitation" should be "hygiene".
Thank you. This has been corrected.
- Discuss why you found majority MB cases. Maybe because weak active case finding in the area which is leading to more severely infected cases/MB.
We have added a discussion of potential reasons for this in discussion.
- Discuss the effect of sampling form local registry because they are programme generated and biased by the local socio-economics and programme operations.
We have edited to make it clear that our samples were convenience samples from local registries.
- inline 202-213, you are not discussing your results. It sounds like background/intro.
We have re-organized our discussion section accordingly.
- inline 214-221 your discussion is good.
Thank you!
- line 226-227 says your study was not designed for co-infection btwn water source, whereas it was your second objective. contradictory statement. you already categorised lep with distance so this is not valid.
We have edited wording for clarity.
- line 246-249 is a repetition. avoid and use space to conclude effectively. you have some good results to conclude, for example, results indicate that poor wash condition is a risk factor for leprosy though association should be further validated with a large sample size study. The Lep and schis co-infection still cannot be ruled out and need stratified analysis with sufficient sample size.
We have shortened the conclusion.
- Based on your results also write a recommendation for programmes in the discussion section.
We have included recommendations that we believe future researchers will find helpful.
Reviewer 2 Report
Results:
1) Line 152 (Table 3): formatting; Variables, 1st row, Age (missing bracket after 18.4)
Discussion:
2) The limitations of this study including its case-control design should be explicitly mentioned within the discussion chapter:
a) that the study can only provide evidence of an association but not a causal relationship
b) the possible influences of the design of the control group to the results of the study should be also discussed
3) Line 223: please explain the presumed cause why there could be an association between schistosomiasis and leprosy within the Lake Tana area
Author Response
Dear IJERPH Editorial Office:
Thank you for giving us the opportunity to revise our draft of the manuscript titled “Poor WASH (Water, Sanitation, and Hygiene) Conditions are Associated with Leprosy in North Gondar, Ethiopia” for publication in the International Journal of Environmental Research and Public Health. We appreciate the reviewers’ feedback and their thoughtful input. We have incorporated most of the reviewers’ suggestions, which are highlighted in the manuscript. Please see our point-by-point response to the reviewers’ comments.
Reviewer 2
1) Line 152 (Table 3): formatting; Variables, 1st row, Age (missing bracket after 18.4)
Discussion:
Thank you. We have corrected this.
2) The limitations of this study including its case-control design should be explicitly mentioned within the discussion chapter:
We have added this in discussion section.
- that the study can only provide evidence of an association but not a causal relationship
We have acknowledged this in the second to last paragraph of the discussion section.
- the possible influences of the design of the control group to the results of the study should be also discussed
This has been addressed in the discussion limitations section.
3) Line 223: please explain the presumed cause why there could be an association between schistosomiasis and leprosy within the Lake Tana area
We have edited accordingly.